# Impact of Endocrine-Disrupting Chemicals in Breast Milk on Postpartum Depression in Korean Mothers

**DOI:** 10.3390/ijerph18094444

**Published:** 2021-04-22

**Authors:** Ju-Hee Kim, Hye-Sook Shin, Woo-Hyoung Lee

**Affiliations:** 1Department of Nursing, College of Nursing Science, Kyung Hee University, Seoul 02447, Korea; suksh@khu.ac.kr; 2Department of Civil, Environmental, and Construction Engineering, University of Central Florida, Orlando, FL 32816, USA; woohyoung.lee@ucf.edu

**Keywords:** postpartum depression, endocrine-disrupting chemicals, breast milk, mono-2-ethylhexyl phthalate, ethyl-paraben

## Abstract

Previous human and animal studies have reported an association between endocrine-disrupting chemicals (EDCs) and anxiety/depression. This study aimed to determine how the concentrations of phthalate metabolites, bisphenol A, triclosan, and parabens in breast milk are associated with the risk of developing postpartum depression (PPD) in Korean mothers. We recruited 221 mothers who were receiving lactation coaching at breastfeeding clinics between July and September 2018. The breast milk samples were collected along with responses to the Edinburgh Postnatal Depression Scale. The multivariable logistic regression results revealed that the phthalate, bisphenol A, parabens, and triclosan levels in the breast milk were not significantly associated with the risk of PPD. This study was the first attempt to analyze the association between the levels of EDCs in breast milk and the risk of PPD. Considering that PPD is a condition that affects not only the women diagnosed with it, but also their children and families, the results of this study may have great relevance to populations in environmentally sensitive periods.

## 1. Introduction

Postpartum depression (PPD) is defined as the occurrence of major depressive episodes either in pregnancy or within the first 4 weeks postpartum, and it affects 7–20% of all mothers [1,2,3,4]. When a woman experiences childbirth, the placenta, which secretes reproductive hormones such as estradiol and progesterone, is also delivered. As a result, women experience marked changes in reproductive hormone levels in the pre- and post-pregnancy and delivery periods, and fluctuations in these reproductive hormones are known to contribute to the development of PPD [2,3,5]. In particular, estradiol increases the cAMP response element-binding protein (CREB) activity and neurotrophin receptor protein trkA, and it decreases GSK-3 beta activity. Progesterone also regulates the synthesis, release, and movement of neurotransmitters [6,7]. These hormones are associated with antidepressive effects and directly or indirectly affect emotions, cognition, and motivation, including the development of PPD [3,8].

The endocrine-disrupting chemicals (EDCs) such as phthalate and bisphenol A (BPA) are synthetic xenoestrogen that mimic, block, and interfere with hormones, for human metabolism, growth, and development, and they have adverse effects on the reproductive system [9,10,11]. At the cellular level, they may inhibit the functions of lysosomes and mitochondria, causing DNA damage and UVB-induced damage through the production of reactive oxygen species (ROS) and nitric oxide (NO) [10]. Anxiety- and depression-related behavioral disorders can occur if EDCs induce estrogen effects on the hippocampus and amygdala, which regulate the hypothalamic–pituitary–adrenal axis (HPA), or influence changes in neurotransmitters that control brain organs [12,13,14]. Phthalates, BPA, triclosan (TCS), and parabens are well-known ubiquitous EDCs and can act like reproductive hormones in the human endocrine system. Phthalate is a material that gives flexibility and durability to plastic, it is used in various personal care products (PCPs) such as food containers, perfumes, and toys, and it is known to affect the human endocrine system, neurodevelopment, as well as the reproductive system [15,16]. BPA, an estrogen-agonist compound, is used in various ways as a manufacture polymer such as in food and beverage containers, dental sealants, and baby bottles [17,18]. Parabens and TCS, which have estrogenic effects, are also used in many PCPs such as cosmetics, soaps, and deodorants [19,20]. The biological mechanism linking EDCs and PPD is still unclear, but several human studies [15,16,17,21] and animal studies [14,15] have reported an association between EDCs and anxiety/depression. Lee et al. [16] analyzed the association between urinary phthalate levels and depressive symptoms in 535 elderly people, and they confirmed that di-2-ethylhexyl phthalate (DEHP) levels had a positive association with depression (affective and spiritual symptoms) with an odds ratio of 1.92. Xu et al. [14,15] reported that DEHP and BPA are associated with anxiety and depressive behavior in mice. This suggests that the exposure to EDCs during vulnerable times, such as during pregnancy and childbirth, also affects the incidence of PPD.

Phthalate, BPA, TCS, and parabens are nonpersistent EDCs and are mainly excreted through urine with a short half-life of 6–29 h [18,22,23,24,25]. However, EDCs with a high log octanol-water partition coefficient (Kow) value tend to be more likely to bioaccumulate because of their low affinity for water [26,27]. Previous studies have reported that some EDCs are metabolized and accumulate in the human body, and they have been detected in serum, hair, and even breast milk and placenta [10,22,26,28]. Biomonitoring is a way of assessing the risk of EDCs, and urine is often tested because it may be acquired noninvasively [29,30]. However, urine samples are known to have high intra- and inter-variability [22,30,31]. A breast milk sample is considered a better target biological matrix than a urine sample, as it is possible to assess the mother’s exposure information during a woman’s life cycle, as well as the newborn’s recent exposure [26,32,33]. In addition, although the bioaccumulation of EDCs occur at a low dose, if exposure occurs repeatedly during an environmentally vulnerable period of an individual’s life, large amounts of EDCs can accumulate in their body [26]. Several studies have demonstrated that prenatal exposure to low doses of BPA (<50 mg/kg/d) affects sex differentiation and anxiety-related behavior [13,34,35,36,37]. Consequently, we hypothesized that the higher the concentration of EDCs in breast milk, the higher the risk of PPD due to increased estrogen effects. Therefore, this study aimed to determine if the concentrations of phthalate metabolites, BPA, TCS, and parabens in breast milk are associated with the risk of developing PPD.

## 2. Materials and Methods

### 2.1. Study Population and Sample Collection

We enrolled all 221 individuals who participated in the Endocrine Disruptors Project for Mothers, which prospectively studied the association between the lifestyle of Korean postpartum women and 15 toxic chemicals in the breast milk and urine for this study [10,22]. To collect a representative sample, we divided Korea into four regions (Seoul metropolitan, Chungcheong, Honam, and Yeongnam region), and we then extracted participants based on the average fertility rate of women of childbearing age for the 3 years (2015–2017) in each region. These 221 mothers were receiving lactation coaching at breastfeeding clinics between July and September 2018. The inclusion criteria were as follows: (1) primiparous mothers, based on a previous study, which showed that the concentration of toxic chemicals in breast milk was different between primiparous and multiparous mothers [38]; (2) mothers who spend the most time with their infants during the day; (3) mothers who were currently breastfeeding; (4) mothers who did not have a history of or were not diagnosed with depression; (5) mothers who had lived in their current residence for more than one year; and (6) mothers who understood the study purpose and agreed in writing to participate. We excluded mothers who had inflammation (such as mastitis), smoking habits, and psychiatric disorders (a condition identified as a mental illness by a mental health professional). However, because we collected data from mothers at breastfeeding clinics in the community, we did not have access to their official medical records and we collected data based on the mothers’ self-reports. A qualified nurse collected 20 mL of hand-expressed breast milk based on a previous study in which the phthalate concentration of breast milk expressed through a machine was higher than that of breast milk expressed by hand [39]. The breast milk samples were then stored in polypropylene tubes that did not contain phthalate and BPA. This study was approved by the Institutional Review Board at Kyung Hee University, Seoul, Korea (KHSIRB-18-029). Written informed consent was obtained from all subjects before participation.

### 2.2. Postpartum Depression Scale

We used the Edinburgh Postnatal Depression Scale (EPDS) developed by Cox, Holden, and Sagovsky [40] to measure PPD. This is a self-reported 10-item scale that has been used to measure PPD in many countries around the world [41]. Participants select the closest of four responses to the emotion they have experienced within the last 7 days, and the score range is 0 to 30. An EPDS score of 9 or more indicates that consultation from an expert is required. Cronbach’s alpha coefficient of the EPDS scores was 0.89 in this study.

### 2.3. Chemical Analysis

We investigated 10 phthalate metabolites (mono-(2-ethyl-5-hydroxyhexyl) phthalate (MEHHP), mono-(2-ethyl-5-oxohexyl) phthalate (MEOHP), mono-(2-ethyl-5-carboxypentyl) phthalate (MECPP), mono-N-butyl phthalate (MnBP), mono-benzyl phthalate (MBzP), mono-(carboxyoctyl) phthalate (MCOP), mono-isobutyl phthalate (MiBP), mono-isononyl phthalate (MiNP), mono-2-ethylhexyl phthalate (MEHP), and monoethyl phthalate (MEP)), BPA, methylparaben (MP), ethylparaben (EP), propylparaben (PP), and TCS in breast milk. We conducted the analysis according to the guidelines of previous studies [42,43], and the detailed analysis steps were described in our previous study [10].

We followed a quality assurance protocol by measuring blank samples and performing internal quality controls for each batch of breast milk samples. For the internal quality control, we evaluated the linearity, accuracy, precision, and detection limits. The calibration curve consisted of five points that covered the concentration range in pooled breast milk; R^2^ was >0.99 in the linearity tests of all analytes. For the accuracy test, we evaluated the recovery rate at each of the three concentration levels (low, medium, and high) by spiking reference materials to the pooled breast milk sample. The recovery percentages were 93–115% in all analytes. For the precision test, we compared inter- and intra-day samples. The maximum value of the relative standard deviation (RSD) was ≤10.0% for both the intra- and inter-day tests of all analytes. The limits of detection (LODs) were as follows: MEHP, 0.139 µg/L; MEP, 0.131 µg/L; MiBP, 0.188 µg/L; MiNP, 0.043 µg/L; MnBP, 0.282 µg/L; MBzP, 0.082 µg/L; MEHHP, 0.139 µg/L; MEOHP, 0.154 µg/L; MECPP, 0.113 µg/L; MCOP, 0.040 µg/L; BPA, 0.076 µg/L; TCS, 0.035 µg/L; EP, 0.035 µg/L; MP, 0.101 µg/L; and PP, 0.134 µg/L. For the external quality control, we sent 30 samples to other laboratories to be analyzed following the same standard operating procedure using LC-MS/MS, with all compounds showing ≥0.8 intra-class correlation values, except for one compound (MnBP; ICC value = 0.53; *p* = 0.054).

### 2.4. Statistical Analysis

For undetected samples below the LOD, we used a proxy value LOD divided by the square-root of 2 [44]. We used log-transformed chemical concentrations due to a skewed distribution of breast milk data. The characteristics of the participants are presented using descriptive statistics, including the frequency, percentage, mean, and standard deviation. Postpartum depression was divided into a PPD group and a non-PPD group based on the cutoff score of 9 points in the EPDS, and the difference between the two groups was verified by the *t*-test, chi-squared test, and Mann–Whitney U test. We performed binary logistic regressions to confirm the associations between EDC concentrations and PPD, and we conducted multiple regression analysis for EDCs that were detected in >70% of participants (MnBP, MEHP, MiNP, and EP). Maternal age, pre-pregnancy body mass index, and household monthly income were adjusted based on a previous study [45]. The significance levels for the statistical testing were set to α = 0.05 and 0.1. Statistical analyses were performed using SAS 9.4 (SAS Institute Inc., Cary, NC, USA).

## 3. Results

### 3.1. Characteristics of the Population

The mean maternal age was 31 years (median, 31 years; range, 19–42 years), with a maternal pre-pregnancy BMI ranging between 16.0 and 33.4 kg/m^2^ (mean, 21.36 kg/m^2^; median, 20.6 kg/m^2^). Most women had an irregular menstrual cycle (65.2%) and menstrual pain (83.7%). The neonatal mean age, birth weight, and birth height values were 34 days (range 3–108 days), 3.2 kg (range 2.2–4.3 kg), and 50.4 cm (range 40–57 cm), respectively (Table 1). When comparing the group with more than 9 and less than 9 points based on the EPDS cutoff, there were differences in maternal age (*p* = 0.045), household income (*p* = 0.030), and EPDS score (*p* < 0.001) between the two groups. The mean EPDS value was 9.11 (SD = 4.3) and the median value was 9.0 (range, 3.0–25.0; Table 1).

### 3.2. Levels of Chemicals in the Breast Milk

Phthalate metabolites (MEP, MnBP, MiBP, MBzP, MEHP, and MiNP) were detected in 5.4–83.3% of the samples, with geometric mean concentrations of 0.04–1.44 μg/L. MEHHP, MEOHP, MECPP, and MCOP were not detected in the breast milk in this study. BPA, TCS, MP, EP, and PP were detected in 25.8–88.2% of the samples, with geometric means of 0.12–0.46 μg/L (Table 2).

MnBP, MEHP, MiNP, and EP were detected in over 70% of samples and were compared between the PPD group and the non-PPD group; the mean value of MEHP (*p* = 0.076) was significantly different between the two groups (Table 3 and Figure 1). The EDCs we tested for in the breast milk samples were highly positively correlated with one another (Appendix A).

The white boxes represent the non-postpartum depression group (*n* = 126), and the gray boxes represent the postpartum depression group (*n* = 95). Boxes represent values between the 25th and 75th percentiles. Black lines indicate the median values, and the whiskers indicate the range of nonoutlier data points. The difference between the two groups was verified by the *t*-test, chi-squared test, and Mann–Whitney U test.

### 3.3. Association of Chemicals with Postpartum Depression

The logistic regression results indicated no association between EDCs and PPD at the significance level *p* = 0.05 (Table 3 and Table 4). However, as the concentration of MEHP in breast milk increased, the odds ratio for PPD occurrence was 1.13 in the group with 9 points or more in the EPDS compared with that in the group with 9 points or less in the EPDS (CI = 1.03–1.12 and *p* = 0.053). Furthermore, as the concentration of EP in breast milk increased, the odds ratio for PPD occurrence was 1.17 in the group with 9 points or more in the EPDS compared with that in the group with 9 points or less in the EPDS (CI = 0.99–1.32 and *p* = 0.091; Table 4).

## 4. Discussion

The occurrence of PPD can be influenced by a variety of factors: biological factors such as reproductive hormones (estradiol and progesterone), lifestyle habits (smoking and sleep), and medical history (depression); social and psychological factors (social support and marital satisfaction); and environmental factors (EDCs and air pollution) [3,8]. In particular, fluctuations in reproductive hormones before and after delivery contribute to the triggering of PPD by altering various biological systems (HPA axis, lactogenic hormone, and the immune system) in hormone-sensitive women [8]. In this study, we investigated the association between PPD and EDC levels in the breast milk of Korean mothers and found no association between PPD and EDC levels. To our knowledge, this is the first study that attempted to determine the association between the risk of developing PPD and the concentration of EDCs in breast milk.

In this study, as the concentration of MEHP in breast milk increased, the odds ratio for the occurrence of PPD was 1.13 in the group with 9 points or more in the EPDS compared to that in the group with 9 points or less in the EPDS (*p* = 0.053), a significance level of 0.1. These results are similar to those of previous studies [14,16]. Lee et al. [16] analyzed the association between urinary phthalate concentrations and depressive symptoms in 535 elderly individuals living in the community, and they confirmed that urinary DEHP levels were positively associated with the risk of depressive symptoms with an odds ratio of 1.92. An animal study also reported that DEHP down-regulates the expression of ER-β in the hippocampus and decreases the levels of p extracellular signal-regulated kinase (p-ERK) 1/2 [14]. This suggests that exposure to DEHP influences anxiety/depression-related behaviors. However, the results of this study are not consistent with several human studies [15,21]. These human studies used US National Health and Nutrition Examination Survey data to analyze the association between urinary phthalate concentrations and depression symptoms in adults, in general, and specifically in the elderly, respectively [15,21]. The former study reported that the urinary concentrations of MECPP, MnBP, MiBP, and MBzP were associated with the depression score [21], and the latter study reported that the concentrations of MCPP, MCNP, and MnBP were related to the depression score [15]. In addition, a recent study on the association between urinary phthalate concentration and PPD symptoms found that the urinary DnOP concentration was 1.48 times higher in patients in the PPD group than in patients in the non-PPD group [46]. In this study, the concentrations of MnBP and MiNP were not significantly associated with PPD. This discrepancy among studies may be due to differences in biological samples and the population. In particular, it is known that the concentration of EDCs in urine is higher than that of breast milk [26]. In this study, the detection rates and GM values of EDCs in urine were 65–100% and 0.66–65.6 μg/g, respectively, whereas the detection rates and GM values of EDCs in breast milk were 5.4–88.2% and 0.04–1.44 μg/L, respectively, in the same population [10,22]. The possible mechanisms underlying the anti-androgen and estrogenic effects of phthalates have been reported in previous studies [14,16]. An experimental study reported an increased expression of ERβ in the hippocampus and decreased levels of p-ERK 1/2 in association with DEHP exposure [14]. Exposure to DEHP leads to oxidative stress and mitochondrial dysfunction in cells, resulting in a disturbance of neurotransmission [47,48,49]. In addition, phthalate exposure can alter lipid metabolism in the brain through the peroxisome proliferator-activated receptor [50], disturb dopamine receptor D2 and the homeostasis of calcium-dependent neurotransmitters [51,52], and affect the biosynthesis and transformation of thyroid hormones [53]. This indicates that phthalates may cause PPD by causing changes in the neurotransmitters of the limbic body.

In this study, BPA was detected in breast milk, but we could not analyze the association with PPD, due to the low detection rate of 48.4%. A recent study on the association between urinary BPA concentration and PPD symptoms reported no association [46]. However, an animal study reported that BPA down-regulates the lacta α-amino-3-hydroxy-5-methyl-4-isoxazolepropionic acid (AMPA) receptor in the amygdala and hippocampus, which is associated with an aggravated anxiety- and depression-like state in mice [13]. Perera et al. [17] confirmed through a cohort study that exposure to prenatal BPA in the third trimester had a positive association with depression and anxiety in boys aged 10–12 years. Several animal experimental studies also reported that prenatal BPA-induced DNA methylation in the BDNF gene altered the expression of genes encoding the estrogen-related receptors γ, ER-α, and ER-β in the cortex and hypothalamus [13,54]. Neurotransmitters such as NMDA receptors and AMPA receptors are associated with antidepressive effects, and BPA down-regulates the AMPA receptor subunit GluR1 in the cortex and hypothalamus [13]. In addition, BPA is known to interfere with gonadal steroid-induced synaptogenesis, resulting in the loss of spine synapses [54]. This indicates that BPA, an estrogen-agonist, directly or indirectly affects the neurotransmitters or genes of the hippocampus and amygdala, resulting in PPD [55].

In this study, as the concentration of EP in breast milk increased, the odds ratio for PPD occurrence was 1.17 in the group with 9 points or more in the EPDS compared with that in the group with 9 points or less in the EPDS (*p* = 0.091) at a significance level of 0.1. Paraben is converted to p-hydroxybenzoic acid after being absorbed by the body and has estrogenic and anti-androgenic effects, inhibits sulfotransferase enzymes, and has a role in testosterone-induced transcriptional inhibition [11,56,57]. Previous studies reported that parabens are associated with oxidative stress, sperm DNA damage, and thyroid hormones [56,58]. This suggests that parabens may cause PPD by similar mechanisms to those observed with other EDCs. The EP concentration in Koreans is relatively high compared to that in other populations, due to fermented foods and relatively relaxed environmental policies [22,47,59,60]. Thus, further studies are needed on the mechanisms of paraben and the effects on negative health including PPD. In this study, we were unable to analyze the association between PPD risk and TCS levels, due to a lower detection rate of 25.8%. TCS, which is logKow 4.76, is also an EDC, and a recent review reported that TCS separates mitochondria and interferes with ion channels even at low doses [19,20,61,62,63]. Thus, EDCs mimic natural hormones, causing the body to over-react to stimuli or respond at inappropriate times and block the effects of hormones on specific receptors in the human endocrine system. They may also stimulate or suppress the endocrine system, causing the over- or under-production of hormones, thereby disturbing the human endocrine system. Consequently, even low doses of EDCs in a vulnerable period of someone’s life can cause PPD if they are subjected to repeated exposure [9,64].

There were some limitations to this study. First, as phthalate, BPA, triclosan, and the parabens analyzed in this study are nonpersistent substances, the results of this study did not include the cumulative exposure according to the life cycle of women, but rather the relationship between temporary exposure and PPD. Second, we analyzed the 1:1 association between phthalate, BPA, triclosan, and parabens and PPD. However, in real life, as exposures to these substances may occur at the same time, the results of this study may be under- or over-interpreted. Third, the health effects of EDCs may vary depending on the dose, frequency of exposure, exposure route, and genetic characteristics of the exposed population [9]. In several studies, estrogen receptor alpha gene (ER-α) has been shown to be a major factor influencing hormonal changes during pregnancy and childbirth [2]. However, we did not consider all of these variables. Lastly, this study investigated the association between EDCs and PPD, one of the environmental factors among the many risk factors for PPD. We presented only the possibility that EDCs could be a small part of the trigger for PPD; thus, further studies with diverse population groups and more sophisticated research designs, including additional variables, are needed to elucidate a causal relationship. Nevertheless, this study attempted to analyze the association between phthalate, parabens, and PPD, and the association between PPD and the concentrations of MEHP and EP were significant when using a significance level of 0.1. Considering that the level of EDCs in breast milk is less than that of urine by 10–20 times [18,23,26], these results are remarkable. This study indicates the potential of EDCs to contribute to PPD.

## 5. Conclusions

This study was conducted to determine how the concentrations of phthalate metabolites, BPA, TCS, and parabens in breast milk were associated with the risk of PPD. We found that no EDCs in the milk were significantly associated with the risk of developing PPD. However, the concentrations of MEHP and EP in breast milk were positively associated with the risk of PPD in primiparous populations of Korean mothers, at a significance level of 0.1. This study was the first to attempt to analyze the association between levels of EDCs in breast milk and the risk of PPD. Considering that PPD affects not only the women diagnosed with the condition, but also their children and families [2,3], the results of this study may have great relevance to populations in environmentally sensitive periods.

## Figures and Tables

**Figure 1 ijerph-18-04444-f001:**
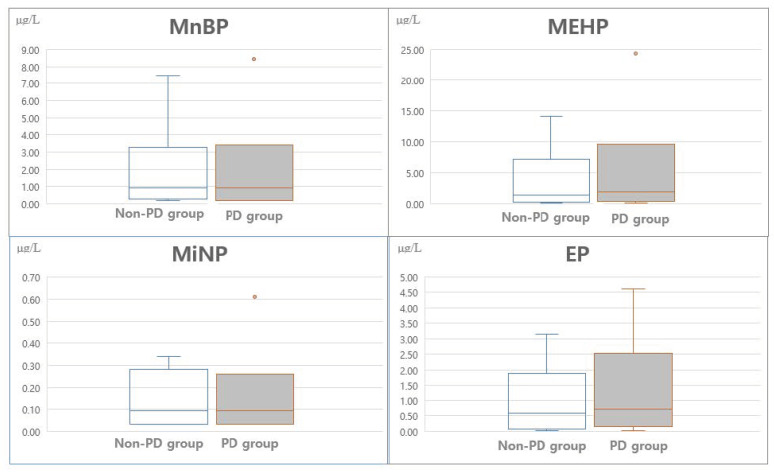
Concentrations of MnBP, MEHP, MiNP, and EP according to the postpartum groups. MnBP, mono-N-butyl phthalate; MiNP, mono-isononyl phthalate; MEHP, mono (2-ethylhexyl) phthalate; EP, ethyl-paraben.

**Table 1 ijerph-18-04444-t001:** Socioeconomic and perinatal characteristics of participants.

Characteristics	Categories	Total	Non-PPD Group (*n* = 126)	PPD Group (*n* = 95)	t/x^2^	*p*
		*n* (%) or M (SD)				
Maternal age (years)		31.3 (3.4)	31.7 (3.6)	30.7 (4.1)	1.93	0.045
Pre-pregnancy BMI (kg/m^2^)		21.3 (3.1)	21.3 (3.3)	21.4 (2.9)	−0.35	0.726
weight (kg)		61.8 (9.4)	62.0 (9.3)	61.5 (9.6)	0.45	0.652
Education	<College	24 (10.6)	6 (4.8)	18 (18.9)	1.94	0.160
	≥College	197 (89.4)	120 (95.2)	77 (81.1)		
Household income (USD/month)	<5000	124 (56.1)	66 (52.4)	58 (61.1)	0.63	0.030
	≥5000	97 (43.9)	60 (47.6)	37 (38.9)		
Employment status	Working	167 (75.6)	97 (77.0)	70 (73.7)	0.26	0.613
	Not working	54 (24.4)	29 (23.0)	25 (26.3)		
Age of first menarche		13.1 (1.4)	13.2 (1.3)	13.0 (1.3)	1.18	0.241
Menstruation cycle	Regular	77 (34.8)	45 (35.7)	32 (33.7)	0.09	0.764
	Irregular	144 (65.2)	81 (64.3)	63 (66.3)		
Menstrual pain	Yes	185 (83.7)	104 (82.5)	81 (85.3)	0.29	0.593
	No	36 (16.3)	22 (17.5)	14 (14.7)		
Neonatal gender	Male	108 (48.9)	57 (45.2)	51 (53.7)	1.24	0.266
	Female	113 (51.1)	69 (54.8)	44 (46.3)		
Neonatal age (day)		33.7 (25.9)	32.7 (25.5)	35.0 (26.6)	−0.64	0.123
Neonatal birth weight (kg)		3.2 (0.4)	3.3 (0.4)	3.20 (0.5)	1.12	0.262
Neonatal birth height (cm)		50.4 (2.3)	50.3 (2.3)	50.6 (2.3)	−1.08	0.283
Neonatal weight (kg)		4.4 (1.2)	4.40(1.16)	4.36 (1.2)	0.25	0.802
Neonatal height (cm)		55.3 (4.8)	55.5 (4.9)	54.9 (4.6)	0.63	0.530
EPDS score		9.1 (4.3)	4.0 (2.7)	12.8 (3.4)	−20.94	<0.001

BMI, body mass index; PPD, postpartum depression; EPDS, Edinburgh Postpartum Depression Scale.

**Table 2 ijerph-18-04444-t002:** Concentrations of phthalate metabolites, bisphenol A, triclosan, and parabens in the breast milk.

Analyte				Percentile (μg/L)				
	LOD	*n* (% > LOD)	GM (SD)	5th	25th	50th	75th	95th
MEP	0.131	102 (46.2)	0.17 (2.37)	<LOD	<LOD	<LOD	0.27	1.96
MnBP	0.282	161 (72.9)	0.83 (3.16)	<LOD	<LOD	0.89	1.86	8.46
MiBP	0.188	153 (69.2)	0.47 (3.19)	<LOD	<LOD	0.49	0.93	7.44
MBzP	0.082	12 (5.4)	0.06 (1.37)	<LOD	<LOD	<LOD	<LOD	0.10
MEHP	0.139	184 (83.3)	1.44 (6.00)	<LOD	0.35	1.72	4.88	24.38
MiNP	0.043	158 (71.5)	0.10 (2.84)	<LOD	<LOD	0.08	0.22	0.61
BPA	0.076	107 (48.4)	0.12 (2.91)	<LOD	<LOD	<LOD	0.23	0.88
TCS	0.035	57 (25.8)	0.04 (2.62)	<LOD	<LOD	<LOD	0.04	0.29
MP	0.101	130 (58.8)	0.33 (5.61)	<LOD	<LOD	0.18	1.12	10.34
EP	0.035	195 (88.2)	0.46 (5.23)	<LOD	0.14	0.62	1.62	4.60
PP	0.134	93 (42.1)	0.21 (3.30)	<LOD	<LOD	<LOD	0.39	2.17

LOD, level of detection; GM, geometric mean; SD, standard deviation; MEP, mono ethyl phthalate; MnBP, mono-N-butyl phthalate; MiBP, mono-isobutyl phthalate; MBzP, monobenzyl phthalate; MiNP, mono-isononyl phthalate; MEHP, mono (2-ethylhexyl) phthalate; BPA, bisphenol A; TCS, triclosan; MP, methyl paraben; EP, ethyl-paraben; PP, propyl paraben.

**Table 3 ijerph-18-04444-t003:** Mean concentrations of MnBP, MEHP, MiNP, and EP according to postpartum groups.

Characteristics	Total	Non-PPD Group	PPD Group	t/x^2^	*p*
		Mean (SD)			
MnBP	0.83 (3.16)	0.81 (2.05))	0.85 (2.96)	1.35	0.178
MEHP	1.44 (6.00)	1.05 (4.86)	1.82 (6.11)	1.79	0.076
MiNP	0.10 (2.84)	0.04 (2.16)	0.15 (2.37)	2.56	0.112
EP	0.46 (5.23)	0.44 (3.96)	0.48 (4.98)	−0.62	0.534

MnBP, mono-N-butyl phthalate; MiNP, mono-isononyl phthalate; MEHP, mono (2-ethylhexyl) phthalate; EP, ethyl-paraben; PPD, postpartum depression; SD, standard deviation.

**Table 4 ijerph-18-04444-t004:** Results of the logistic regression for the association between chemical concentrations and the EPDS.

	Crude OR	95% CI	*p*	Adjusted OR	95% CI	*p*
MnBP	0.95	0.61–1.37	0.541	0.94	0.64–1.38	0.745
MEHP	1.12	1.04–1.14	0.050	1.13	1.03–1.12	0.053
MiNP	0.22	0.44–1.38	0.330	0.23	0.04–1.35	0.394
EP	1.18	0.99–1.40	0.089	1.17	0.99–1.32	0.091

Regression models are adjusted for age, pre-pregnancy body mass index, and household monthly income. Reference values: EPDS cutoff score <9. OR, odds ratio; CI, confidence interval; MnBP, mono-N-butyl phthalate; MiNP, mono-isononyl phthalate; MEHP, mono (2-ethylhexyl) phthalate; EP, ethyl-paraben.

## Data Availability

Restrictions apply to the availability of these data.

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
