# Peer review of "Impact of Endocrine-Disrupting Chemicals in Breast Milk on Postpartum Depression in Korean Mothers"

_ijerph, 2021, doi:10.3390/ijerph18094444_

Round 1
Reviewer 1 Report
The article was corrected properly and will now be a solid article for the journal. Furthermore, the writing is better than many other published articles that I have seen in many other journals. It is a good article that I look forward to citing in my own future research.
Author Response
Thank you for reviewing our manuscript again and for your positive feedback.
Reviewer 2 Report
The authors answered all the comments and the manuscript has been improved for publication.
Author Response

(The authors gave the same response as above.)

Reviewer 3 Report
'However, because we collected data from mothers at breastfeeding clinics in the community, we did not check their official medical records and collected data based on the mothers' self-reports'.
This needs to be included in the manuscript when describing the inclusion/exclusion criteria.
Author Response
Thank you for reviewing our manuscript again and for your comment. In accordance with your suggestion, we have added this to the description of inclusion/exclusion criteria. Please see page 3, lines, 103-105 of the revised manuscript.
Reviewer 4 Report
As is well known that postpartum depression (PPD) is a serious and common psychiatric disorder and was reported with association with many endocrine-disrupting chemicals (EDCs). Kim et al carried out an interesting bimornitoring investigation on the association of PPD with the exposure to EDCs. In total 221 Korean mothers were recruited and Phthalate metabolites (MEP, MnBP, MiBP, MBzP, MEHP, and MiNP), bisphenol A, triclosan, and parabens were detected in breast milk. No significant association was revealed between these EDCs and PPD risk, despite that the concentrations of MEHP and EP was positively associated with the risk of PPD at a significance level of 0.1. Genrally speaking, this is a careful study. The experiments were design properly and the results were solid. Therefore, I recommend this manuscript to be published in International Journal of Environmental Research and Public Health after revision following listed issues:
Specific comments for authors:
- EDCs are chemicals that mimic, block, or interfere with hormones in the body's endocrine system. Line 45, for the description of EDCs, it would be better to revise it, “EDCs act as an estrogen receptor agonist, an antagonist of 45 the androgen receptor”. Actually the antaogonist of estrogen receptor and agonist/antagonist of estorgen receptor also belong to EDCs.
- MnBP, MEHP, MiNP, and EP were detected in over 70% of samples. Actually many studies reported the association of some EDCS with PPD. A latest study on the prenatal exposure to bisphenols and phthalates was reported with PPD (Prenatal exposure to bisphenols and phthalates and postpartum depression: The role of neurosteroid hormone disruption.The Journal of Clinical Endocrinology & Metabolism. org/10.1210/clinem/dgab199). It would be better to make a brief discussion.
- In the abstract, the authors stated that ‘the concentrations of mono-2-ethylhexyl phthalate (MEHP; p =.053) and ethyl-paraben (EP; p =.091) in breast milk were positively associated with the risk of PPD at a significance level of 0.1. ’ Statistically speaking, p > 0.05 means no significant.
- As for the sampled participants, more general description would be desired and relevant discussion would be welcome. Is the number of participants is engough for such investigation.?
- As for the statistis, such as logistic regression, although the p values are above 0.05, it can still deliver some useful information.
- For line 308, ‘Nevertheless, this study was the first attempt to 308 analyze the association between phthalate, parabens, and PPD, ‘. It would be better to delete the word ‘the first’. Similar case in line 320.
- Please double check the whole manuscript for potential grammar errors and typos.
Author Response
Specific comments for authors:
Thank you for reviewing our manuscript and for your positive feedback.
- EDCs are chemicals that mimic, block, or interfere with hormones in the body's endocrine system. Line 45, for the description of EDCs, it would be better to revise it, “EDCs act as an estrogen receptor agonist, an antagonist of 45 the androgen receptor”. Actually the antaogonist of estrogen receptor and agonist/antagonist of estorgen receptor also belong to EDCs.
Response 1:
Thank you for your comments. In accordance with your suggestion, we have revised the text. Please see page 1, lines 40-43, of the revised manuscript.
“The endocrine disrupting chemicals (EDCs) such as phthalate and bisphenol A (BPA) are synthetic xenoestrogen that mimic, block, and interfere with hormones, for human metabolism, growth and development, and has adverse effects on the reproductive system”
- MnBP, MEHP, MiNP, and EP were detected in over 70% of samples. Actually many studies reported the association of some EDCS with PPD. A latest study on the prenatal exposure to bisphenols and phthalates was reported with PPD (Prenatal exposure to bisphenols and phthalates and postpartum depression: The role of neurosteroid hormone disruption. The Journal of Clinical Endocrinology & Metabolism.org/10.1210/clinem/dgab199). It would be better to make a brief discussion.
Response 2:
Thank you for providing this reference. We have referred to it in the Discussion section and added it as a reference. Please see page 8, lines 244-246, and page 9, lines, 264-265, of the revised manuscript.
“In addition, a recent study on the association between urinary phthalate concentration and PPD symptoms reported that the urinary DnOP concentration in PPD patients was 1.48 times higher in patients in the PPD group than in patients in the non-PPD group [53].”
“A recent study on the association between urinary BPA concentration and PPD symptoms reported no association [53].”
- In the abstract, the authors stated that ‘the concentrations of mono-2-ethylhexyl phthalate (MEHP; p =.053) and ethyl-paraben (EP; p =.091) in breast milk were positively associated with the risk of PPD at a significance level of 0.1. ’ Statistically speaking, p > 0.05 means no significant.
Response 3:
We agree with your opinion. In accordance with your suggestion, we have revised the wording. Please see page 1, lines 18-20, of the revised manuscript.
“The multivariable logistic regression results revealed that phthalate, bisphenol A, parabens, and triclosan levels in the breast milk were not significantly associated with the risk of PPD”
- As for the sampled participants, more general description would be desired and relevant discussion would be welcome. Is the number of participants is enough for such investigation?
Response 4:
In accordance with your suggestion, we have added details on the sampling of participants. Please see page 2, lines, 87-93, of the revised manuscript.
“We enrolled all 221 individuals who participated in the Endocrine Disruptors Project for Mothers, which prospectively studied the association between the lifestyle of Korean postpartum women and 15 toxic chemicals in the breast milk and urine for this study [22,10]. To collect a representative sample, we divided all regions of Korea into four regions (Seoul metropolitan, Chungcheong, Honam, and Yeongnam region), and then extracted participatns based on average fertility rate of women of childbearing age for the 3-year (2015-2017) in each region”
- As for the statistics, such as logistic regression, although the p values are above 0.05, it can still deliver some useful information.
Response 5:
We agree with you. We changed the wording in the Statistical analysis section. Please see page 4, lines 157-158, in the revised manuscript.
“The significance level for the statistical testing was set to α = 0.05 and α = 0.1.”
- For line 308, ‘Nevertheless, this study was the first attempt to 308 analyze the association between phthalate, parabens, and PPD, ‘. It would be better to delete the word ‘the first’. Similar case in line 320.
Response 6:
In accordance with your suggestion, we have deleted “the first.” Please see page 10, lines 316-318, of the revised manuscript.
“Nevertheless, this study was attempt to analyze the association between phthalate, parabens, and PPD, and association between PPD and concentrations of MEHP and EP were significant when using a significance level of 0.1.”
- Please double check the whole manuscript for potential grammar errors and typos.
Response 7:
In accordance with your suggestion, we had the manuscript proofread by a professional editing company.

This manuscript is a resubmission of an earlier submission. The following is a list of the peer review reports and author responses from that submission.
Round 1
Reviewer 1 Report
To my knowledge, it is interesting and important issues for the environmental epidemiology before I carefully read it. Although they did a lot of statistics, the associations between PPD and EDCs are not significant. I cannot find the notable and significant findings in this manuscript. The evidence is not enough to support the hypothesis.
Author Response
Response
Thank you for your valuable comments and suggestion. We have revised this paper while considering your comments.

Reviewer 2 Report
I think that the article was well-written and presented a very good argument that links PPD with these particular EDCs. I would only add in a bit more of a discussion about possible cross-correlation during the linear regression. For example, there could be more about were the 10 phthalate metabolites showing cross-correlation with each other, or with BPA (for example)?
Author Response
I think that the article was well-written and presented a very good argument that links PPD with these particular EDCs. I would only add in a bit more of a discussion about possible cross-correlation during the linear regression. For example, there could be more about were the 10 phthalate metabolites showing cross-correlation with each other, or with BPA (for example)?
Response
Thank you for the valuable input. We added the results of the suggested correlation and description in Supplementary Table 1 and the Results section. Please see page 6 (lines 192-194) in the revised manuscript and the supplementary file.
“The EDCs we tested for in the breast milk samples were highly positively correlated with one another (Supplementary table 1).
Supplementary table 1. Spearman correlations among phthalates, BPA, triclosan, and paraben in breast milk
|
Chemicals |
MEP |
MnBP |
MiBP |
MEHP |
MiNP |
MBzP |
BPA |
TCS |
MP |
EP |
PP |
|
|
MEP |
1.000 |
0.919** |
0.630** |
0.016 |
0.021 |
0.116 |
0.029 |
-0.030 |
0.026 |
-0.082 |
-0.010 |
|
|
MnBP |
  |
1.000 |
0.663** |
0.142* |
0.138* |
0.257** |
0.086 |
-0.034 |
0.109 |
0.006 |
0.076 |
|
|
MiBP |
  |
  |
1.000 |
0.356** |
0.684** |
0.445** |
0.262** |
-0.030 |
0.748** |
0.274** |
0.682** |
|
|
MEHP |
  |
  |
  |
1.000 |
0.637** |
0.382** |
0.205** |
-0.027 |
0.554** |
0.233** |
0.589** |
|
|
MiNP |
  |
  |
  |
  |
1.000 |
0.540** |
0.343** |
-0.019 |
0.947** |
0.405** |
0.927** |
|
|
MBzP |
  |
  |
  |
  |
  |
1.000 |
0.262** |
-0.023 |
0.554** |
0.292** |
0.520** |
|
|
BPA |
  |
  |
  |
  |
  |
  |
1.000 |
0.277** |
0.371** |
0.399** |
0.476** |
|
|
TCS |
  |
  |
  |
  |
  |
  |
  |
1.000 |
0.000 |
0.139* |
0.098 |
|
|
MP |
  |
  |
  |
  |
  |
  |
  |
  |
1.000 |
0.444** |
0.959** |
|
|
EP |
  |
  |
  |
  |
  |
  |
  |
  |
  |
1.000 |
0.499** |
|
|
PP |
  |
  |
  |
  |
  |
  |
  |
  |
  |
  |
1.000 |
|
MEP, mono ethyl phthalate; MnBP, mono-N-butyl phthalate; MiBP, mono-isobutyl phthalate; MBzP, monobenzyl phthalate; MiNP, mono-isononyl phthalate; MEHP, mono (2-ethylhexyl) phthalate; BPA, bisphenol A; TCS, triclosan; MP, methyl paraben; EP, ethyl paraben; PP, propyl paraben; * p < 0.05, ** p < 0.01

Reviewer 3 Report
The aim of this study was to determine the effect of chemicals in breast milk on the risk of developing postpartum depression in Koreans mothers. The findings are interesting and the design of this study was well done. However, I have critical comments for this manuscript.
The title should be changed as the authors did not demonstrate
that postpartum depression and endocrine disrupting of breast milk
Authors should be more general such as
“Impact of chemicals in breast milk on the postpartum depression in Korean mothers”
Line 19-20. No significant association was detected. Authors should change this sentence for:
concentrations of MEHP and EP in breast milk tended to be positively correlated with the risk of PPD after …
line 74. The breast milk is not a “biomarker” that is discharged from the breast. Please correct this sentence/word. Example: Breast milk is produced by women’s breast and EDCs accumulate well…
Line 140. Change “maternal” for “Maternal”.
Legend for figure 1 should describe the statistical analysis used and the sample size.
A Figure with the regression linear of (a) the PPD and MEHP conc and (b) PPD and EP conc would be more relevant for this study than the figure 1.
Authors must change the sentence 196-197. PPD and concentrations of MEHP and EP tended to be positively correlated (due to p = 0.053 and p = 0.091). These associations were not significant
Line 268. No significant association was detected. Authors should change this sentence for:
concentrations of MEHP and EP in breast milk tended to be positively correlated with the risk of PPD in …
Author Response
The aim of this study was to determine the effect of chemicals in breast milk on the risk of developing postpartum depression in Koreans mothers. The findings are interesting and the design of this study was well done. However, I have critical comments for this manuscript.
- The title should be changed as the authors did not demonstrate
that postpartum depression and endocrine disrupting of breast milk
Authors should be more general such as
“Impact of chemicals in breast milk on the postpartum depression in Korean mothers”
Response
Thank you for your valuable suggestion. In response to your comment, we changed the title as follows. Please see page 1 of the revised manuscript.
“Impact of endocrine disrupting chemicals in breast milk on postpartum depression in Korean mothers”
- Line 19-20. No significant association was detected. Authors should change this sentence for:
concentrations of MEHP and EP in breast milk tended to be positively correlated with the risk of PPD after …
Response
In response to your comment, we revised that sentence as follows. Please see the Abstract, page 1 (lines 18-22) of the revised manuscript
“Multiple logistic regression results demonstrated that no EDCs in the milk were significantly associated with PPD risk. However, the concentrations of mono-2-ethylhexyl phthalate (MEHP; p =.053) and ethyl-paraben (EP; p =.091) in breast milk were positively associated with the risk of PPD after adjusting for maternal age, pre-pregnancy body mass index, and household monthly income) at a significance level of 0.1.”
- line 74. The breast milk is not a “biomarker” that is discharged from the breast. Please correct this sentence/word. Example: Breast milk is produced by women’s breast and EDCs accumulate well…
Response
In response to your comment, we revised that sentence as follows. Please see the Introduction section, page 2 (lines 84-85) of the revised manuscript
“Breast milk is produced in the women’s breast, and EDCs accumulate well in the breast adipose tissue”
- Line 140. Change “maternal” for “Maternal”.
Response
In response to your comment, we revised maternal to Maternal (Materials and methods section, page 4, line 155).
- Legend for figure 1 should describe the statistical analysis used and the sample size.
Response
In response to your comment, we added the statistical analysis used and the sample size as follows. Please see the Results section, page 7 (lines 206-209) of the revised manuscript
“The white boxes represent the non-postpartum depression group (n=126), and the gray box represents the postpartum depression group (n=95). Boxes represent values between the 25th and 75th percentiles. Black lines indicate the median values, and the whiskers indicate the range of non-outlier data points. The difference between the two groups was verified by t-test, chi-squared test, and Mann-Whitney U test.”
- A Figure with the regression linear of (a) the PPD and MEHP conc and (b) PPD and EP conc would be more relevant for this study than the figure 1.
Response
Thank you for your valuable suggestion. We presented the comparison of the EDCs concentrations between the PPD and the non-PPD groups in Table 3; however, for the reader's understanding, we presented figure 1 additionally. We thought that the regression results presented in Table 4 are clearer than those presented in the figure. We will consider the regression model picture in future studies.
- Authors must change the sentence 196-197. PPD and concentrations of MEHP and EP tended to be positively correlated (due to p = 0.053 and p = 0.091). These associations were not significant
Response
In response to your comment, we revised that sentence as follows. Please see the Results section, page 7 (lines 210-211) of the revised manuscript.
“The logistic regression results indicated no association between EDCs and PPD at the significance level p= 0.05 (Tables 3 and 4).”
- Line 268. No significant association was detected. Authors should change this sentence for:
concentrations of MEHP and EP in breast milk tended to be positively correlated with the risk of PPD in …
Response
We agree with your valuable comment. Considering the low level of EDCs in the breast milk, this study set the significance levels at 0.1 and 0.05. As you said, we have added a sentence indicating that there was no association between EDCs and PPD, throughout the paper (Abstract (page 1, lines 18-19), Result (page 7, lines 210-211), Discussion (page 8, line 238 and page 9, 289), Conclusion (page 10, lines 328-331)). In addition, in order to convey the research results clearly, we stated that the association between MEHP, EP, and PDD was slightly related at the significance level of 0.1. Please see pages 8 (line 238) and 9 (line 289) of the revised manuscript.
“We found that no EDCs in the milk were significantly associated with the risk of developing PPD. However, the concentrations of MEHP and EP in breast milk were positively associated with the risk of PPD in primiparous populations of Korean mothers, at a significance level of 0.1.”

Reviewer 4 Report
In this study, the authors explore the association between EDCs and PPD. Several major points need to be addressed:
Introduction
line 43: A more detailed description should be given for EDCs (i.e. what type of compounds they are).
line 51: Additional information shoulf be provided for the particular ECRs that were analyzed in this study.
Materials and Methods
2.1: What about history of psychiatric disorders in the women included in the study, in their partners and their family? This is a very important confounding factor which should be in detail addressed and factored in in the data analysis as a parameter. The inclusion/exclusion criteria should be also mentioned in detail here, not only referring the reader to a previous study.
Also lifestyle habits, i.e. smoking, living in a urban vs. rural environment, chronic disorders are significant confounding factors and should be taken into account in data analysis. Otherwise, the merit of the results is low.
Discussion
The authors should highlight what is new in this study compared to their previous studies.
PPD studies addressing biological fluids with holistic approaches and correlated to PPD symptom severity should be also discussed.
Author Response
In this study, the authors explore the association between EDCs and PPD. Several major points need to be addressed:
- Introduction
1-1. line 43: A more detailed description should be given for EDCs (i.e. what type of compounds they are).
Response
Thank you for your valuable suggestion. In response to your comment, we added more detailed description. Please see the Introduction section, page 2 (lines 43-46) of the revised manuscript.
“The endocrine disrupting chemicals (EDCs) such as phthalate and bisphenol A (BPA), are synthetic xenoestrogen that disrupt the endocrine system, releasing specific hormones necessary for human metabolism, growth and development, and has adverse effect on the reproductive system”
1-2. line 51: Additional information should be provided for the particular ECRs that were analyzed in this study.
Response
Thank you for your valuable suggestion. In response to your comment, we added more detailed description. Please see the Introduction section, page 2 (lines 53-62) of the revised manuscript.
“Phthalates, BPA, triclosan (TCS), and parabens are well-known ubiquitous EDCs and can act like reproductive hormones in the human endocrine system. Phthalate is a material that gives flexibility and durability to plastic and is used in various personal care products (PCPs) such as food containers, perfumes, and toys, and is known to affect the human endocrine system, neurodevelopment, as well as the reproductive system. BPA, an estrogen-agonist compound, is used in various ways as a manufacture polymer such as in food and beverage containers, dental sealants, and baby bottles. Parabens and TCS, which have estrogenic effects, are also used in many PCPs such as cosmetics, soaps, and deodorants”
- Materials and Methods
What about history of psychiatric disorders in the women included in the study, in their partners and their family? This is a very important confounding factor which should be in detail addressed and factored in in the data analysis as a parameter. The inclusion/exclusion criteria should be also mentioned in detail here, not only referring the reader to a previous study. Also lifestyle habits, i.e. smoking, living in a urban vs. rural environment, chronic disorders are significant confounding factors and should be taken into account in data analysis. Otherwise, the merit of the results is low.
Response
We agree with your opinion. We excluded from this study mothers with chronic diseases that could affect the development of depression or the EDCs concentration in breast milk; only healthy mothers were recruited. In response to your comment, we added the inclusion/exclusion criteria details in the methods section. Please see pages 2-3 (lines 97-103) of the revised manuscript.
“The inclusion criteria were as follows: 1) primiparous mothers, based on a previous study, which showed that the concentration of toxic chemicals in breast milk was different between primiparous and multiparous mothers; 2) mothers who spend the most time with their infants during the day; 3) mothers who were currently breastfeeding; 4) mothers who had lived in their current residence for more than one year; and 5) mothers who understood the study purpose and agreed in writing to participate. We excluded mothers who had inflammation, such as mastitis, psychiatric disorder, and smoking habits.”
- Discussion
3.1. The authors should highlight what is new in this study compared to their previous studies.
Response
Thank you for your valuable suggestion. This study was the first attempt to analyze the association between EDCs and PPD, and the direct comparison was difficult because there was no previous study. However, we have written the discussion section with reference to the association between EDCs and depression. In response to your comment, we indicated the highlights of this study in the discussion section. Please see pages 8-10 (lines 233-235 and lines 322-324) of the revised manuscript.
“Nevertheless, this study was the first attempt to analyze the association between phthalate, paraben, and PPD, and we found the potential of EDCs to contribute to PPD”
3.2. PPD studies addressing biological fluids with holistic approaches and correlated to PPD symptom severity should be also discussed.
Response
Thank you for your valuable suggestion. We agree with your opinion. In addition to EDCs, the occurrence of PPD can be influenced by a variety of personal diet and lifestyle, genetic and environmental factors. In response to your comment, we have added this as the limitation of the study (page 10, lines 317-322).
“Lastly, the occurrence of PPD can be influenced by a variety of factors including the personal diet and lifestyle habits, genetic factors, and environmental factors. However, when analyzing the association between EDCs and PPD, we only considered age, BMI, and economic status. Further studies are needed with more diverse population groups, and more sophisticated research designs, including additional variables.”
Round 2
Reviewer 3 Report
The authors corrected appropriately all the comments. Thank you.
Author Response
I appreciate your opinions.
Reviewer 4 Report
I thank the authors for their answers to my comments. However, some issues were not addressed:
- Family/partner history of psychiatric disorders is very important and the authors have not addressed my question concerning this matter. How family/partner history of psychiatric diagnoses was factored in in the analysis?
- Also concerning the exclusion of psychiatric disorder diagnosis. What do the authors mean by psychiatric disorder? This needs to be clarified. Were women with depression diagnosis also excluded?
- The authors have not addressed my comment on 3.2. The authors should include in their discussion other studies on PPD which use biological fluids and address lifestyle factors and symptom severity and discuss them in comparison with their results.
Author Response
Reviewer #4
I thank the authors for their answers to my comments. However, some issues were not addressed:
- Family/partner history of psychiatric disorders is very important and the authors have not addressed my question concerning this matter. How family/partner history of psychiatric diagnoses was factored in in the analysis? 2. Also concerning the exclusion of psychiatric disorder diagnosis. What do the authors mean by psychiatric disorder? This needs to be clarified. Were women with depression diagnosis also excluded?
Response 1&2:
We apologize for not having answered to Reviewer #4's comments satisfactorily. We excluded mothers who had a history of or were diagnosed with depression or other psychiatric conditions from this study. However, because we collected data from mothers at breastfeeding clinics in the community, we did not check their official medical records and collected data based on the mothers' self-reports.
In this study, psychiatric disorder was defined as a condition identified as a mental illness by a mental health professional. In response to your comment, we have added the following information in the inclusion/exclusion criteria on page 3, lines 101, 105-106 of the revised manuscript:
“4) mothers who did not have a history of or were not diagnosed with depression,”
“We excluded mothers who had inflammation (such as mastitis), smoking habits, and psychiatric disorders (a condition identified as a mental illness by a mental health professional).”
- The authors have not addressed my comment on 3.2. The authors should include in their discussion other studies on PPD which use biological fluids and address lifestyle factors and symptom severity and discuss them in comparison with their results.
Response 3:
We apologize for not having answered Reviewer #4's comments satisfactorily.. We agree that various factors including biological fluids, lifestyle, and severity of depression can influence the occurrence of PPD. We tried to control for the confounding variables as much as possible. We investigated only an association between PPD and EDC levels among the many PPD risk factors and added the following content in the first section of the Discussion on page 8, lines 231-237, and in the limitations section of Discussion on page 10, lines 231-237, 326-331 of the revised manuscript:
“The occurrence of PPD can be influenced by a variety of factors: biological factors such as reproductive hormones (estradiol and progesterone), lifestyle habits (smoking, sleep), and medical history (depression); social and psychological factors (social support, marital satisfaction); and environmental factors (EDCs, air pollution) [3, 8]. In particular, fluctuations of reproductive hormones before and after delivery contribute to triggering PPD by altering various biological systems (HPA axis, lactogenic hormone, and the immune system) in hormone-sensitive women [8]. In this study, we investigated an association between PPD and EDC levels in the breast milk of Korean mothers by adjusting for age, pre-pregnancy body mass index, and household monthly income.”
“Lastly, this study investigated the association between EDCs and PPD, one of the environmental factors among the many risk factors for PPD. We presented only the possibility that EDCs could be a small part of the trigger for PPD, thus, further studies with diverse population groups and more sophisticated research designs, including additional variables, are needed to elucidate a causal relationship.”
